# Investigation on Influence Factors of Photo-Induced PLZT-Based Ion Drag Pump

**DOI:** 10.3390/mi15121424

**Published:** 2024-11-27

**Authors:** Xinjie Wang, Zhen Lv, Yuming Shao, Yujie Shi, Yao Yao, Jiong Wang

**Affiliations:** School of Mechanical Engineering, Nanjing University of Science and Technology, Nanjing 210094, China; zlv@njust.edu.cn (Z.L.); ymshao@njust.edu.cn (Y.S.); yjshi@njust.edu.cn (Y.S.); yyao@njust.edu.cn (Y.Y.); wjiongz@njust.edu.cn (J.W.)

**Keywords:** electrohydrodynamics, PLZT ceramic, charge injection, ion drag pumps, pumping performance

## Abstract

The ion drag pump, as one kind of electrohydrodynamic pump, has received considerable attention in fluid applications due to its excellent pumping flow rate and pressure. However, there is a lack of systematic research about the factors that influence pumping performance of the ion drag pump. Here, a photo-induced ion drag pump based on the PLZT ceramic is proposed by combining the photoelectric effect and field emission phenomenon. The EHD model of this ion drag pump is constructed based on the mathematical model of the photovoltage of the PLZT ceramic, through which a series of finite element simulations are carried out to comprehensively investigate the factors that influence the pumping performance. The results demonstrate that such an ion drag pump is able to be improved by optimizing the electrode structure and fluid channel; increasing the light intensity; and providing a basic design guideline for applications of ion drag pumps in microfluidics, soft robots, and heat dissipation in micro devices.

## 1. Introduction

In order to satisfy the fluid control in specific application scenarios, such as microfluidic systems, and heat dissipation in microelectromechanical systems (MEMS), lots of microscale pumping components have been developed, principally using piezoelectric- [1,2], electrostatic- [3,4], pneumatic- [5,6,7], and electromagnetic-driven [8,9] mechanisms; other reported mechanisms include three electrohydrodynamic (EHD) pumping counterparts (i.e., ion drag [10], conduction [11,12], and induction pumps [13]). By distinguishing the presence or absence of moving parts in the pump, former driven mechanisms are classified as mechanical pumps, and EHD pumps are classified as non-mechanical pumps [14]. Compared to mechanical pumping, these EHD pumps are able to directly transfer electrical energy into fluid kinetic energy, thus possessing distinct advantages, such as an absence of vibration and noise, a simple structure, excellent stability, and reliability due to their having no moving parts. Therefore, EHD pumping exhibits promising prospects in fluid driving fields, especially microfluidics, soft robots, and heat dissipation in micro devices [15,16].

According to different mechanisms of charge generation, there are basically three kinds of EHD pumps: ion drag pumping (also called charge injection pumping), conduction pumping, and induction pumping, respectively [17]. Ion drag pumps rely on charges generated by charge injection around the electrode–fluid interface; these generated charges are directionally actuated by the Coulomb force, thus leading to fluid flow. Conduction pumps utilize charges generated by a non-equilibrium relationship of dissociation-recombination of molecules to realize fluid pumping. In conduction pumps, heterocharge layers are formed near the electrodes as a result of the chargeability of electrodes, which are characterized by a high concentration of ions with opposite polarity with respect to the nearest electrode [16]. As a result, charges in fluid are attracted to electrodes with opposite polarity (i.e., causing fluid flow). Moreover, charges are generated in induction pumping owing to a gradient of electric conductivity in fluid, consequently causing fluid flow [18]. Among those three EHD pumps, especially, ion drag pumping is widely developed due to its excellent pumping flow rate and pressure [10,19].

To date, numerous studies have been conducted on ion drag pumping, involving numerical simulation, experimental testing, and applications. Since Sharbaugh investigated the pumping of transformer oil through an ion drag pump consisting of a closed loop of glass tubing in 1985 [20], a great deal of experimental research involving ion drag pumping has been studied. For instance, a micro ion drag pump with planar electrodes on a glass substrate was easily fabricated and experimentally tested, showing a quadratic increase of flow rate and pressure with the increase of applied voltage [21]. Seo et al. systematically reported a planar ion drag pump with five kinds of improved electrode configuration. The results showed that ion drag pumps with sawtooth and arc electrodes lead to better pumping performance [22]. Recently, Cacucciolo et al. fabricated stretchable pumps and fiber pumps using the charge injection. The two developed components both show much potential in wearable haptics and thermoregulatory textiles [16,23]. Moreover, the pumping characteristics of ion drag pumps have also been investigated using numerical simulations. Nishikawara et al. studied ion drag pumps with one centrally located emitter and two collector electrodes. The simulations indicate that collector electrodes with a triangular cross-section yield better performance than those having rectangular cross-sections [24]. Using the lattice Boltzmann method, Gao et al. reported three flow modes in the ion drag pump with different electrode arrangements. They revealed that the geometric configurations of the planar interdigitated electrodes have direct effects on the pumping performance [25,26]. In conclusion, research about ion drag pumping is focused on experiments and the mechanism of simulations. However, a few analyses have been conducted to systematically study the factors influencing the pumping performance of the ion drag pump.

Here, we proposed a photo-induced ion-drag-pump-based PLZT ceramic by combining photoelectric effect and charge injection. The EHD model for the proposed photo-induced ion drag pump is constructed. Then, a series of finite element simulations are conducted to systematically study the effect of pumping performance of the photo-induced ion drag pump, involving light intensity, electrode configurations, channel dimensions, and fluid viscosity.

## 2. Materials and Methods

### 2.1. Working Mechanisms of Photo-Induced PLZT-Based Ion Drag Pump

Figure 1a shows the conceptual schematic of the PLZT-based ion drag pump, which consists of a PLZT ceramic served as the energy device and an ion drag pump chip. These two parts are electrically connected via wires. The ion drag pump chip is designed as a sandwich-like multilayer structure (i.e., the substrate layer, the channel layer, and the cover layer), shown in Figure 1b. Note that the interdigitated electrode structure is patterned on the substrate layer to realize an expected electric field, which actuates the dielectric fluid flowing in the channel layer. In order to avoid the dielectric breakdown, the cover layer and the substrate layer are supposed to be strictly dielectric. For this design, when the PLZT ceramic is exposed to UV light with a near 365 nm wavelength, a photovoltage appears between the two electrodes of the PLZT ceramic and increases rapidly because of the photoelectric effect. Then, the photovoltage generates a photoelectric field between the interdigitated electrodes. As shown in Figure 2, once the photoelectric field is high enough to overcome the energy barrier, field emission occurs and the electrons are injected from the cathode into the dielectric liquid in the flow channel. Here, the cathode and anode are regarded as the emitter and collector, respectively. Then, the generated electrons are adhered on neutral molecules, causing these molecules to become ions. With the electric field between the interdigitated electrodes, these resulting ions can be actuated to the anode and discharged, dragging neutral fluid and thus leading to fluid flow. After the UV light is removed, the fluid flow in the channel stops because of the absence of the photovoltage. As discussed above, such a developed photo-induced ion drag pump based on the PLZT ceramic is expected to realize a controllable fluid flow via photovoltage. Furthermore, due to not using external power equipment but the PLZT patch as the energy source, this photo-induced ion drag pump is easy to miniaturize.

### 2.2. Mathematical Modeling of Photo-Induced PLZT-Based Ion Drag Pump

In order to obtain an accurately controllable ion drag pump, the mathematical modeling of the photovoltage is supposed to be achieved. According to previous research, an equivalent electrical model of the PLZT ceramic is shown in Figure 3a. The PLZT ceramic is theoretically equivalent to the parallel connection of a current source *I*_p_, a capacitor *C*_p_, and a resistor *R*_p_. Note that the equivalent electrical model of the PLZT ceramic changes when the PLZT ceramic is connected with external loads (i.e., ion drag pump chip). To achieve an accurate prediction of the photovoltage for the PLZT ceramic, an equivalent electrical model for this photo-induced PLZT-based ion drag pump is constructed by taking external loads into account. In this way, the ion drag pump chip can be regarded as a parallel connection with the PLZT ceramic, as shown in Figure 3b, where *C*_e_ and *R*_e_ are the external load capacitor and resistor, respectively. By combining the proposed equivalent electrical model of the photo-induced ion drag pump and the mathematical model of photovoltage for the PLZT ceramic [27], the photovoltage of the photo-induced ion drag pump during both the light-on and light-off phase can be expressed as follows:(1)$$V\left(t\right)=V_{s}(1-e^{-t/\tau })+C_{1}(1-e^{-t/\tau_{\theta }})$$
(2)$$V_{d}\left(t\right)=V_{0}+C_{2}e^{-t/\tau_{d}}$$
where *t* is the time; *V*_s_ is the saturated photovoltage; *τ* and *τ*_θ_ are the time constant during the light-on phase and thermal time constant, respectively; *V*_0_ is the photovoltage value after ceasing the UV light; and *C*_1_ and *C*_2_ are the composite coefficient.

Moreover, an electrohydrodynamics (EHD) model for the ion drag mechanism is constructed to understand the fluid-flow behavior, which is subjected to the Navier–Stokes momentum equation:(3)$$\rho \left[\frac{\partial u}{\partial t}+\left(u\cdot \nabla \right)u\right]=-\nabla p+\mu \nabla^{2}u+F_{e}$$
where *u* and *μ* are the flow rate and fluid viscosity, *p* is the pressure, and *F_e_* is the electric body force acting on the fluid. According to the Korteweg–Helmholtz equation, the electric body force *F_e_* can be expressed as follows:(4)$$F_{e}=\rho_{e}E-\frac{1}{2}E^{2}\nabla \varepsilon +\frac{1}{2}\nabla \left[E^{2}\rho {\left(\frac{\partial \varepsilon }{\partial \rho }\right)}_{T}\right]$$
where *ρ_e_*, *E*, *ε*, and *ρ* are the space charge density, electric field, permittivity of the fluid, and density of the fluid, respectively. According to the Poisson’s equation for static electricity, the relationship between the electric field and voltage is:(5)$$\nabla^{2}V\;=\;-\frac{\rho_{e}}{\varepsilon },\;E\;=\;-\nabla V$$

The space charge density *ρ_e_* is subjected to the law of conservation of electric charge:(6)$$\frac{\partial \rho_{e}}{\partial t}+\nabla \cdot J=0$$
where *t* is the time, and *J* is the current density, which represents total amount of charge passing per unit area per unit time. The current density *J* can be expressed as follows:(7)$$J=\mu_{e}\rho_{e}E+D\nabla \rho_{e}+\rho_{e}u+\sigma_{e}E$$
where *μ_e_*, *D*, *u*, and *σ_e_* are the migration coefficient, molecular diffusion coefficient, flow rate, and coefficient of conductivity, respectively.

### 2.3. Characterization Experiments of Photovoltage for PLZT Ceramic

In order to investigate the photovoltage characteristics of the PLZT ceramic, numerous experiments are carried out under different intensities, including 100, 150, and 200 mW/cm^2^. As shown in Figure 4a, when the PLZT ceramic is irradiated by a low light intensity UV light (i.e., 100 mW/cm^2^), a rapidly increasing photovoltage between the two electrodes of the PLZT ceramic occurs and stabilizes over time. After removing the UV light, the generated photovoltage decreases at a relatively slower rate, compared with the increasing rate of the photovoltage. Clearly, high light intensity leads to an increase of the peak photovoltage. However, at high light intensity, such as 150 and 200 mW/cm^2^, a trend of first increasing and then decreasing occurs, as shown in Figure 4a. Figure 4b–d depict the experimental data and the fitting curve of the photovoltage during the light phase and light-stopping phase, and the fitting expressions of photovoltage under the light intensities of 100, 150, and 200 mW/cm^2^ by using Equations (3) and (4) are shown in Figure 4b–d. Moreover, the photovoltage responses of 50 and 300 mW/cm^2^ are also tested in order to obtain the fitting curve between the light intensity and the peak photovoltage, shown in Figure 5, which is used for the EHD simulations in Section 3.

## 3. Results and Discussion

### 3.1. Setup of Simulation for Photo-Induced PLZT-Based Ion Drag Pump

On the basis of the constructed EHD model for the PLZT-based ion drag pump, a finite element simulation can be carried out to study the output performance of this proposed PLZT-based ion drag pump. The setup of the simulation field is illustrated in Figure 6, where the structure parameters of the simulation field are shown in Table 1. It needs to be clarified that the length of channel *l_c_* is changed as a result of the variations of the electrode structure; it is expressed as follows:(8)$$l_{c}=(2\times w_{e}+g_{e})\times N+g_{p}\times (N-1)+2\times l_{e}$$

Figure 6a,b depict the partial enlargement of the ion drag chip structure, comprised of the cross-sectional schematic and top view, respectively. It is noted that an ion injection region is set as the emitter around the negative electrodes to simulate the field emission phenomenon.

Furthermore, in our simulations, the permittivity of the dielectric fluid used in our simulation is regarded as constant, resulting in ∇ε, ∂ε = 0. Consequently, the electric body force *F_e_* acting on the fluid can be expressed as follows:(9)$$F_{e}=\rho_{e}E$$

In other words, the electric body force only includes the Coulomb force. In this way, a greater voltage results in a greater electric body force on the fluid, leading to more efficient pumping. Moreover, the molecule diffusion has little effect on the fluid flow, and the conductive current in the simulation can be ignored because of invariance of the dielectric property of fluid (i.e., *D* = 0, *σ_e_* = 0). In this way, the current density *J* in the Equation (9) is changed:(10)$$J=\mu_{e}\rho_{e}E+\rho_{e}u$$

It should be noted that the electric current produced by the PLZT ceramic (about several nA) is so small compared to the EHD current (about several μA) [22] that the electric current produced by the PLZT ceramic can be ignored in the EHD model.

Finally, a variety of finite element simulations for the PLZT-based ion drag pump are carried out to study the influencing factors of the output performance, such as the intensity of UV light; fluid channel dimensions; fluid properties; and electrode structure. It is clarified that the average flow rate in the outlet of the channel, and the pressure difference between the inlet and outlet of the fluid channel, are, respectively, defined as *u_av_* and *p_d_*, which are used to assess the output performance of the proposed pump.

### 3.2. Effect of the Electrode Structure

Firstly, the effect of the electrode structure on the pumping performance is investigated, including the number of electrode pairs, electrode gap *g_e,_* and electrode pairs gap *g_p_*. Figure 7 shows the output performance of the photo-induced ion drag pump in response to the variation of the number of electrode pairs under different light intensities. Obviously, a higher light intensity leads to a greater photovoltage and consequently induces a larger average flow rate *u_av_* as shown in Figure 7a. Also, as the number of electrode pairs increases, the flow rate u*_av_* increases slowly in a given light intensity, such as from the light intensity of 50 to 300 mW/cm^2^; the average flow rate changes from 1.3 to 4.6 mm/s (about 3.5 times) and by 1.3 times (from 1.3 to 1.7 mm/s) by increasing the number of electrode pairs from 1 to 8 pairs. Similarly, the pressure difference curve has almost the same trend compared to the average flow rate curve. When increasing the light intensity, the flow rate and pressure difference exhibit an upward trend, as shown in Figure 7b.

Next, we investigated the pumping performance as a function of the electrode gap *g_e_*. In a given voltage or light intensity, a smaller electrode gap induces a more effective pumping performance, as shown in Figure 8. This is because a smaller electrode gap leads to a stronger electric field, which consequently increases the electric body force *F_e_* in Equation 11. Figure 8a depicts the flow rate distribution in the outlet of the fluid channel under a light intensity of 100 mW/cm^2^. It is clear that the flow rate around the middle of the channel is maximized and increases when the electrode gap decreases. As a result, a more effective EHD pump could be realized by decreasing the electrode gap. The pressure difference curves under different light intensities and different electrode gaps are shown in Figure 8b.

As shown in Figure 1, there is an opposite electric body force that exists between the electrode pairs, which hinders the EHD flow in the channel. That is why increasing the number of electrode pairs leads to a small improvement in the pump performance. Therefore, the effect of the gap between the electrode pairs should be studied to improve the pumping performance. Figure 9 shows the output performance curve of the EHD pump, where the average flow rate *u_av_* and the pressure difference *p_d_* show an upward and then downward trend with the increasing of the gap between the electrode pairs. Firstly, by increasing the gap between the electrode pairs, a rapid decrease of the electric field between the electrode pairs leads to an efficient ion drag pump because a decrease in the obstruction of EHD flow. Then, a further increase in the gap between the electrode pairs causes an increase in the fluid channel, which in turn results in an increase in the along-travel losses, while other parameters remain unchanged. As a result, further increasing the gap between the electrode pairs causes a decrease in pumping performance. In conclusion, the simulation results in Figure 9 show that the optimum value of the gap between the electrode pairs exists around 2 mm, with the given structure parameters shown in Table 1.

### 3.3. Effect of Fluid Channel and Fluid Property

Furthermore, the effect of the fluid channel on the pumping performance is investigated, including the channel height and channel length. The effect of channel height is illustrated in Figure 10. Observing the pumping performance curve in Figure 10a, it is evident that the flow rate and pressure are significantly relevant to the channel height. It is noted that increasing the channel height results in an increase of average flow rate but a decrease in pressure difference between the inlet and outlet of the channel. By further investigating the flow rate distribution in the outlet of the channel, the maximum of flow rate moved to the side with the electrodes arranged. As a result of the asymmetry of the electrode distribution, the electric field decreases rapidly in the area away from the electrode in the fluid channel and then induces the flow rate distribution shown in Figure 10b. The normalized channel height is defined as the ratio of actual height *h* to channel height *h*_c_, which is equal to 1 when the position is located in the top of the channel outlet. As the fluid channel height increases, the phenomenon of the asymmetry of the flow rate distribution is more visible, as shown in Figure 10b. In conclusion, the EHD flow in our ion drag pump is mainly on the side close to the electrode, which explains, to some extent, the flow characteristics in Figure 9. As the height of the fluid channel decreases, the resistance to flow on the side of the channel with the embedded electrodes increases, leading to a decrease in the flow rate but an increase in pressure.

Figure 11 shows the effect of channel height and fluid viscosity on the pumping performance, respectively. When increasing the channel length, the along-travel loss due to frictional resistance caused by the blocking effect of the channel walls is similarly increased. Therefore, according to Bernoulli’s principle, the flow rate and pressure differences exhibit a downward trend, which can be validated in Figure 10. Finally, the s series of simulations is conducted to study the effect of fluid viscosity under the light intensity of 100 mW/cm^2^. The pumping flow rate and pressure curve are shown in Figure 11. As we can see, the fluid viscosity can greatly affect the pumping characteristics. When the fluid viscosity changes from 1 × 10^−3^ Pa∙s to 1 × 10^−2^ Pa∙s, the average flow rate *u_av_* decreases to 7% (from 208 to 15 mm/s) and the pressure difference *p_d_* changes from 178 to 70 mPa. In this way, the pumping performance can be improved by selecting a kind of dielectric liquid with low viscosity.

In these sections, we have systematically investigated factors that influence pumping performance for our photo-induced EHD pump by a series of simulations. On the basis of these simulations, such a PLZT-induced EHD pump with excellent pumping performance can be designed on purpose, which is promising for microfluidic, heat dissipation applications.

## 4. Conclusions

In this paper, a photo-induced ion drag pump based on PLZT ceramic are proposed by combining the photoelectric effect and field emission phenomenon. On the basis of the mathematical model of photovoltage during both the light-on and light-off phase, the electrohydrodynamics (EHD) model for the photo-induced ion drag pump is constructed. Moreover, a series of finite element simulations are carried out to investigate the factors that influence the pumping performance for this proposed photo-induced EHD pump, including the electrode structure, channel dimensions, and fluid viscosity, which provide a guideline for the design of this pump. In order to improve the pumping performance, several strategies can be conducted, such as decreasing the electrode gap and increasing the ultraviolet light intensity and the number of electrode pairs to a certain extent. Furthermore, an efficient ion drag pump can be obtained by designing the gap between the electrode pairs, and the channel height. Basically, through these simulations, we have demonstrated that such a photo-induced EHD pump is able to realize fluid pumping based on the PLZT ceramic, which exhibits much potential among the microfluidic, heat dissipation applications.

## Figures and Tables

**Figure 1 micromachines-15-01424-f001:**
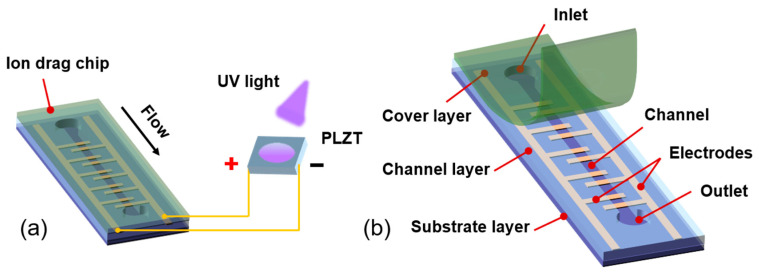
(**a**) Schematic of the photo-induced ion drag pump based on PLZT ceramic; (**b**) the ion drag pump chip with sandwich-like multilayer structures.

**Figure 2 micromachines-15-01424-f002:**
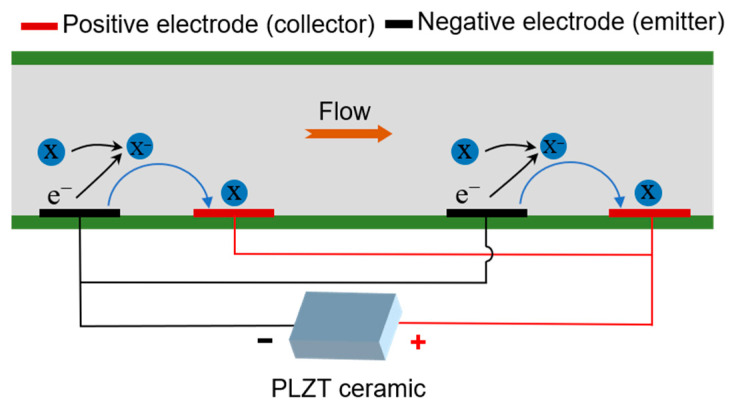
Working mechanism of the photo-induced ion drag pump.

**Figure 3 micromachines-15-01424-f003:**
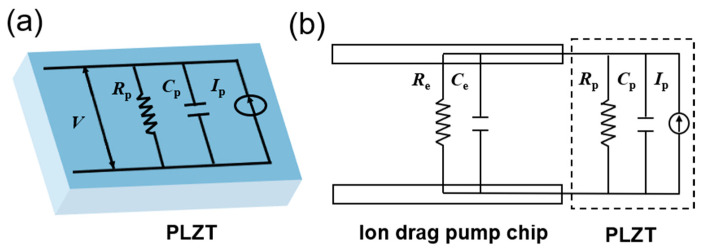
The equivalent electrical model of (**a**) the individual PLZT ceramic and (**b**) the proposed photo-induced ion drag pump.

**Figure 4 micromachines-15-01424-f004:**
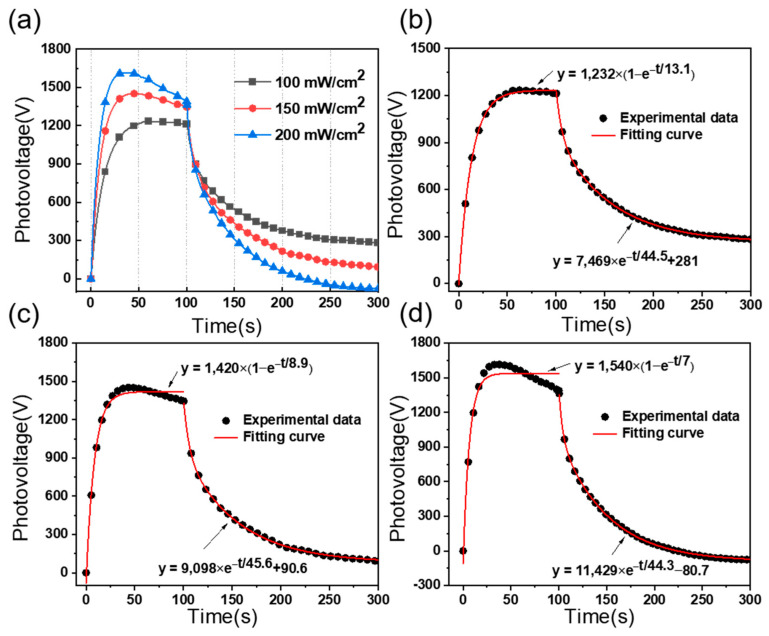
The output characteristics of photovoltage for PLZT ceramic. (**a**) Plot of photovoltage versus time under different UV light intensities. The fitting and experimental curve of photovoltage under UV light intensities of (**b**) 100 mW/cm^2^, (**c**) 150 mW/cm^2^, and (**d**) 200 mW/cm^2^.

**Figure 5 micromachines-15-01424-f005:**
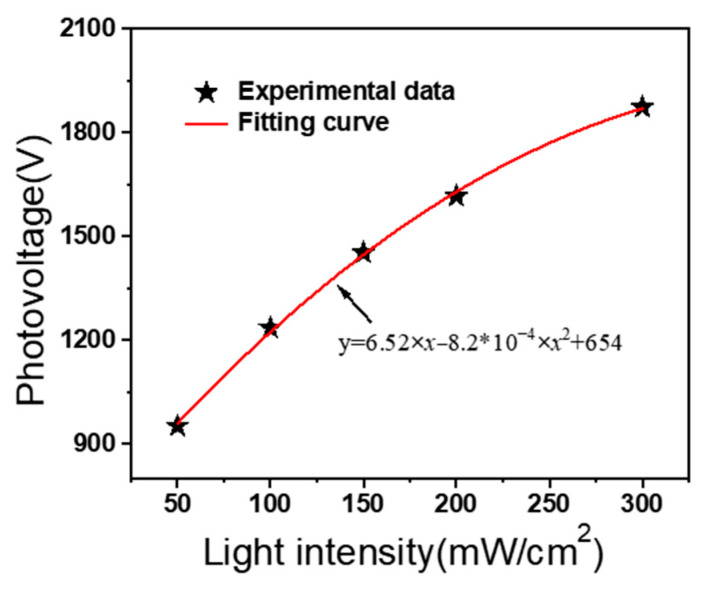
The fitting curve of the maximum photovoltage versus the light intensity.

**Figure 6 micromachines-15-01424-f006:**
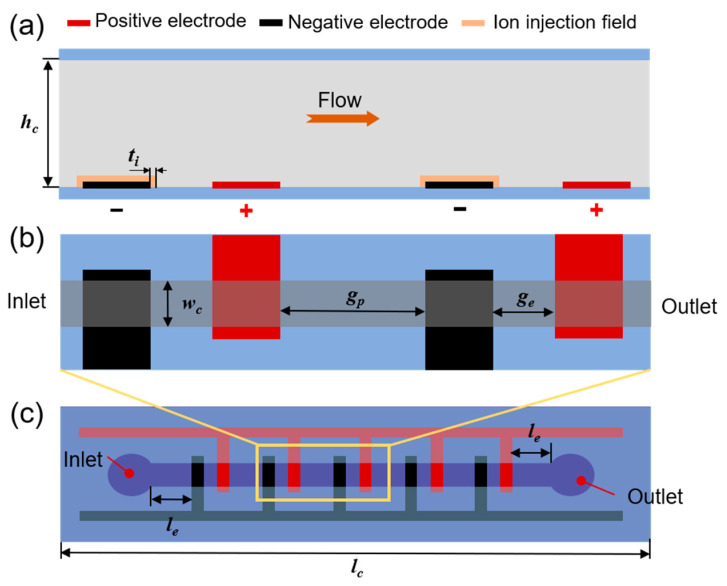
Schematic illustration of simulation field for the ion drag pump. (**a**) Cross-section schematic, (**b**) top view of the finite element simulation field, and (**c**) top view of the proposed EHD chip.

**Figure 7 micromachines-15-01424-f007:**
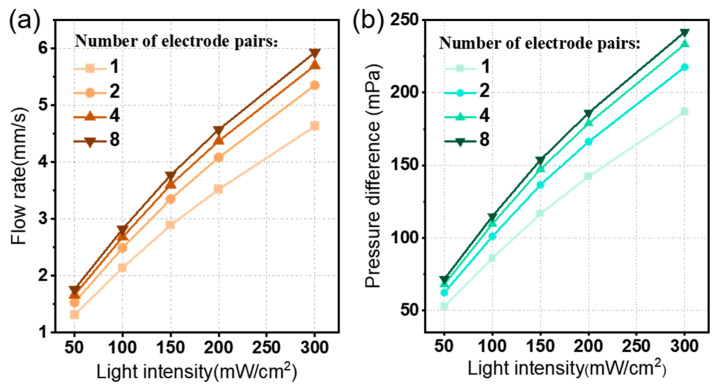
The effect of numbers of electrode pairs on the pumping performance of the photo-induced ion drag pump. Plot of (**a**) the average flow rate *u_av_* and (**b**) pressure difference *p_d_*.

**Figure 8 micromachines-15-01424-f008:**
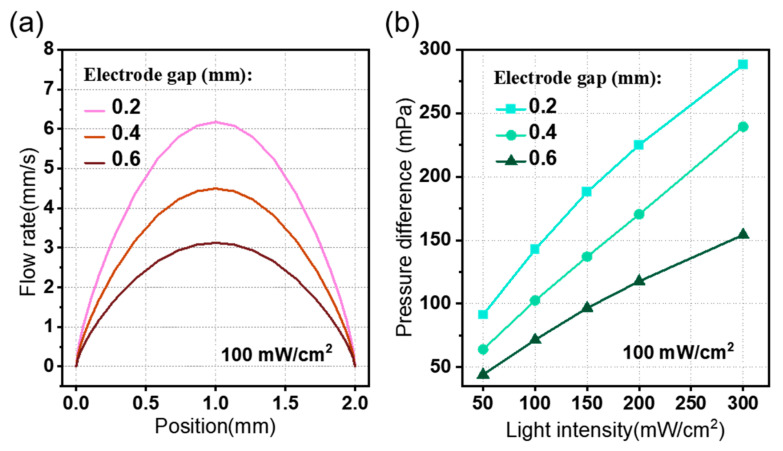
The effect of the electrode gap on the pumping performance of the photo-induced ion drag pump. Plot of (**a**) the average flow rate *u_av_* and (**b**) pressure difference *p_d_*.

**Figure 9 micromachines-15-01424-f009:**
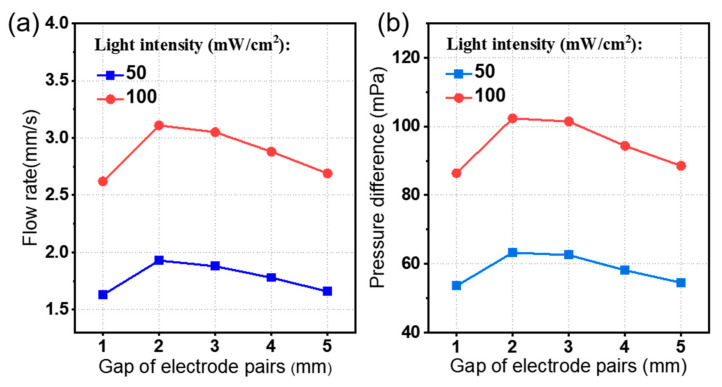
The effect of the gap between the electrode pairs on the pumping performance of the photo-induced ion drag pump. Plot of (**a**) the average flow rate *u_av_* and (**b**) pressure difference *p_d_*.

**Figure 10 micromachines-15-01424-f010:**
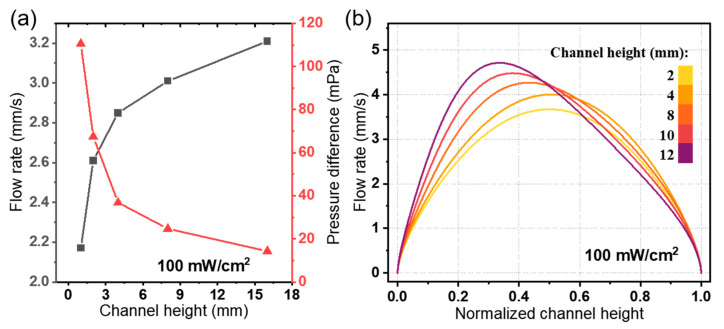
The effect of the channel height on the pumping performance of the photo-induced ion drag pump. (**a**) Plot of the average flow rate *u_av_* and pressure difference *p_d_*; (**b**) the flow rate distribution in the outlet of the channel.

**Figure 11 micromachines-15-01424-f011:**
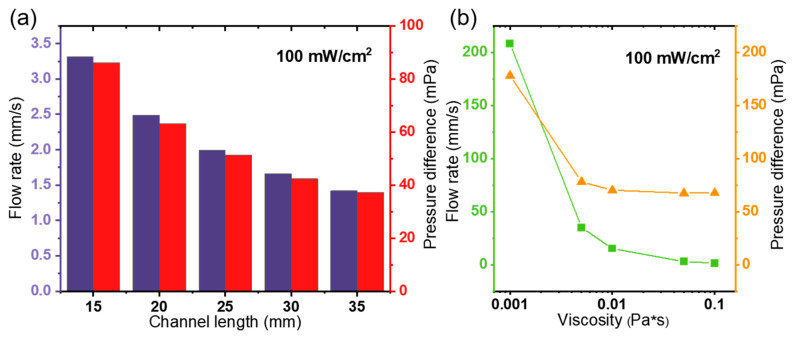
The effect of (**a**) the channel length and (**b**) fluid viscosity on the pumping performance of the photo-induced ion drag pump.

**Table 1 micromachines-15-01424-t001:** The simulation parameters of the ion drag pump.

Parameters	Value
Length of channel *l_c_*	Shown in Equation (10)
Weight of channel *w_c_*	/
Height of channel *h_c_*	2 mm
Weight of electrode *w_e_*	0.5 mm
Gap between positive and negative electrode *g_e_*	0.5 mm
Gap between electrode pairs *g_p_*	1 mm
Thickness of the ion injection field *t_i_*	15 μm
Pairs of electrodes *N*	4
The length of channel end *l_e_*	5 mm

## Data Availability

The original contributions presented in this study are included in the article. Further inquiries can be directed to the corresponding author.

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
