# Peer review of "Investigation on Influence Factors of Photo-Induced PLZT-Based Ion Drag Pump"

_micromachines, 2024, doi:10.3390/mi15121424_

Round 1

Reviewer 1 Report

Comments and Suggestions for Authors

In this paper, a novel photo-induced ion drag pump based on PLZT ceramic is proposed by combining the photoelectric effect and field emission phenomenon. Furthermore, the output performance of this presented ion drag pump is investigated in detail, which provides great potential to facilitate the current research in ion drag pumps. The paper is well organized. But there are also several issues should be addressed:

1)    Figure 10 shows the output performance curve of the EHD pump under different channel height, in which increasing the channel height results in an increase of average flow rate but a decrease in pressure difference. But why? The principles need to be given accordingly in order to explain the phenomenon.

2)    What’s response speed of the photo-induced PLZT-based ion drag pump once the light irradiation is applied to the PLZT ceramaic.

3)    Variables in equations 1 and 2 are undefined in the manuscript, for example, VsV0 and so on.

4)    There are typing errors in Table 1. “Weight of channel” and “Weight of electrode” should be “Width of channel” and “Width of electrode”.  According to Table 1, the value of the width of channel is none, why?

5)    Edition, avoid grammatical mistakes in the submitted manuscript, shown as follow:

“Figure 9 shows….the pressure difference shows an upward and then downward trend ….”

Subscripts of the parameters are sometime italic type and sometime are roman type, e.g. ge and gp

Author Response

In this paper, a novel photo-induced ion drag pump based on PLZT ceramic is proposed by combining the photoelectric effect and field emission phenomenon. Furthermore, the output performance of this presented ion drag pump is investigated in detail, which provides great potential to facilitate the current research in ion drag pumps. The paper is well organized. But there are also several issues should be addressed:

Q1: Figure 10 shows the output performance curve of the EHD pump under different channel height, in which increasing the channel height results in an increase of average flow rate but a decrease in pressure difference. But why? The principles need to be given accordingly in order to explain the phenomenon.

A: Thanks for this helpful suggestion. We are sorry to make the reviewer confused about this problem. Therefore, we have added the fundamental explanations in the revised manuscript for the effect of the channel height to the pumping performance, as follows:

“In conclusion, the EHD flow in our ion drag pump is mainly on the side close to the electrode, which explains, to some extent, the flow characteristics in Figure 9. As the height of the fluid channel decreases, the resistance to flow on the side of the channel with the embedded electrodes increases, leading to a decrease in flow rate but an increase in pressure.”

Q2: What’s response speed of the photo-induced PLZT-based ion drag pump once the light irradiation is applied to the PLZT ceramic.

A: We thank the reviewer for this very relevant comment. For the presented photo-induced PLZT-based ion drag pump, the response speed is subjected to the response speed of the photovoltage, shown in Figure 4 in the submitted paper. Once the photovoltage reaches a certain threshold, the dielectric fluid in the EHD channel will immediately be pumped.

Q3: Variables in equations 1 and 2 are undefined in the manuscript, for example, Vs, V0 and so on.

A: We thank the reviewer for this useful comment. Following the advice, we have defined variables in equations 1 and 2 in the revised manuscript. The revised parts are shown as follows:

“Where t is the time, Vs is the saturated photovoltage, τ and τθ are the time constant during the light-on phase and thermal time constant, respectively, V0 is the photovoltage value after ceasing the UV light, C1 and C2 are composite coefficient.”

Q4: There are typing errors in Table 1. “Weight of channel” and “Weight of electrode” should be “Width of channel” and “Width of electrode”.  According to Table 1, the value of the width of channel is none, why?

A: Thanks for this helpful suggestion. We have modified the errors in Table 1 by using “Width of channel” instead of “Weight of channel”. In the proposed paper, the finite element simulation is carried out on the basis of two-dimensional model. Therefore, the value of the width of channel is none.

Q5: Edition, avoid grammatical mistakes in the submitted manuscript, shown as follow:

“Figure 9 shows the output performance curve of the EHD pump, where the average flow rate and the pressure difference shows an upward and then downward trend with the increasing of the number of electrode pairs.”

Subscripts of the parameters are sometime italic type and sometime are roman type, e.g. ge and gp.

A: We thank the reviewer for this useful comment. The grammatical mistakes in the submitted paper are revised accordingly. We have also carefully gone through the language throughout the manuscript, examining "the taste" of the sentences and modifying them when deemed necessary. Furthermore, we have modified the problem about subscripts of the parameters by checking the submitted paper.

Reviewer 2 Report

Comments and Suggestions for Authors

This paper deals with photo-induced ion drag pumps through utilizing a composite mechanism: photoelectric effect superimposed to the field emission phenomenon. The paper is very interesting and, in my opinion, it would be possible to publish the manuscript in micromachines after solving the following questions.

1) Based on the Figure 9 given in the manuscript, it can be seen that the average flow rate and the pressure difference show an upward and then downward trend with the increasing of the number of electrode pairs. However, there is no fundamental explanation provided in this proposed manuscript.

2) The working voltages in such an ion drag pump are about kilovolts, so how to ensure the safety of this presented pump in applications, such as microfluidic, soft robots and heat dissipation in micro devices?

3) According to the mathematical model of the photo-induced PLZT-based ion-drag pump, it is hard to find the relationship between flow rate and the photovoltage. It’s suggested to add more explanation about this.

4) How about the electric current produced by PLZT ceramic for ion drag pump. According Equations (8), J is the current density. Is there any relation between the photo-induced current and current density J.

5) The title of the Conclusion part is not suitable. The conclusion not only includes the modeling photo-induced PLZT-based ion-drag pump but also the investigation for the influence factors of the pumping performance, and so on.

Author Response

This paper deals with photo-induced ion drag pumps through utilizing a composite mechanism: photoelectric effect superimposed to the field emission phenomenon. The paper is interesting and, in my opinion, it would be possible to publish the manuscript in micromachines after solving the following questions.

Q1: Based on the Figure 9 given in the manuscript, it can be seen that the average flow rate and the pressure difference show an upward and then downward trend with the increasing of the number of electrode pairs. However, there is no fundamental explanation provided in this proposed manuscript.

A: We thank the reviewer for this useful comment. As suggested by the reviewer, we have added the fundamental explanation for the effect of the gap of electrode pairs to the pumping performance, as follows:

“Firstly, by increasing the gap of electrode pairs, a rapid decrease of the electric field between the electrode pairs leads to an efficient ion drag pump because a decrease in the obstruction of EHD flow. Then, a further increase in the gap of electrode pairs causes an increase in the fluid channel, which in turn results in an increase in the along-travel losses, while other parameters remain unchanged. As a result, further increasing the gap of electrode pairs causes a decrease in pumping performance. In conclusion, the simulation results in Figure 9 show that the optimum value of the gap between the electrode pairs exists around 2 mm with given structure parameters in Table 1.”

Q2: The working voltages in such an ion drag pump are about kilovolts, so how to ensure the safety of this presented pump in applications, such as microfluidic, soft robots and heat dissipation in micro devices?

A: We thank the reviewer for this very relevant comment. We acknowledge that the working voltages of the proposed EHD pump are about kilovolts, but would like to emphasize that, compared to the conventional high voltage power supply, the working voltages produced by PLZT ceramics are with ultralow currents (about several nA), which makes the proposed EHD pump more compatible and reliable in practical applications, such as microfluidic, soft robots and heat dissipation in micro devices. Furthermore, we designed a sandwich-like multilayer structure consisted of the substrate layer, electrode layer and cover layer to protect our EHD pumps from the electrostatic breakdown and charge leakage. In this way, the effect of the high working voltages in applications can be controlled to a certain extent.

Q3: According to the mathematical model of the photo-induced PLZT-based ion-drag pump, it is hard to find the relationship between flow rate and the photovoltage. It’s suggested to add more explanation about this.

A: Thanks for the helpful suggestion. As shown in equations (3), (5) and (9), the greater the voltage, the greater the electric field, which in turn leads to a greater electric body force on the fluid. That is, the greater the voltage, the greater the flow rate. The explanations about the relationship between flow rate and the photovoltage are added as follows:

“In other words, the electric body force only includes the Coulomb force. In this way, a greater voltage results in a greater electric body force on the fluid, leading to a more efficient pumping.”

Q4: How about the electric current produced by PLZT ceramic for ion drag pump. According Equations (8), J is the current density. Is there any relation between the photo-induced current and current density J.

A: We thank the reviewer for this useful comment. The electric current produced by PLZT ceramic (about several nA) is so small compared to the EHD current (about several μA) that the electric current produced by PLZT ceramic can be ignored in our EHD mathematical model. As suggested by the reviewer, we have added the explanations about the relationship between the electric current produced by PLZT ceramic and the current density J, as follows:

“It should be noted that the electric current produced by PLZT ceramic (about several nA) is so small compared to the EHD current (about several μA) [22] that the electric current produced by PLZT ceramic can be ignored in the EHD model.”

Q5: The title of the Conclusion part is not suitable. The conclusion not only includes the modeling of the photo-induced PLZT-based ion-drag pump but also the investigation for the influence factors of the pumping performance, and so on.

A: Following the reviewer's advice, we have carefully revised the conclusion part and added some content about the influence factors of the pumping performance, which now reads:

“In this paper, a photo-induced ion drag pump based on PLZT ceramic are proposed by combining the photoelectric effect and field emission phenomenon. On the basis of the mathematical model of photovoltage during both the light-on and light-off phase, the electrohydrodynamics (EHD) model for the photo-induced ion drag pump is constructed. Moreover, a series of finite element simulations are carried out to investigate the influence factors of the pumping performance for this proposed photo-induced EHD pump, including the electrode structure, channel dimensions, fluid viscosity, which provides a guideline for the design of this pump. In order to improve the pumping performance, several strategies can be conducted, such as decreasing the electrode gap, increasing the ultraviolet light intensity and the number of electrode pairs in a certain extent. Furthermore, an efficient ion drag pump can be obtained by designing the gap of electrode pairs and channel height. Basically, through these simulations, we have demonstrated that such a photo-induced EHD pump is able to realize the fluid pumping based on PLZT ceramic, which exhibits much potential among the microfluidic, heat dissipation applications.”

Reviewer 3 Report

Comments and Suggestions for Authors

This research explores a photo-induced ion drag pump based on PLZT ceramic, integrating the photoelectric effect and field emission phenomena, to enhance electrohydrodynamic fluid pumping. Through mathematical modeling and finite element simulations, it systematically investigates the effects of design parameters like electrode configurations, light intensity, and fluid properties on pumping performance, offering valuable guidelines for applications in microfluidics, soft robotics, and heat dissipation technologies. Here are my comments:

The paper relies entirely on simulations. Where is your contribution?

You need provide the real photo of the pump in experiment and experiment setup environment.

You explored several configurations for electrodes. How did you decide the optimal gap and number of electrode pairs? Could there be configurations that you have not yet tested?

The manuscript lacks an in-depth investigation into temperature effects and material degradation of PLZT ceramics under UV light.

This paper lacks a recent related advancements, like Multimodal Strain Sensing System for Shape Recognition of Tensegrity Structures by Combining Traditional Regression and Deep Learning Approaches; and Predictive modeling of flexible EHD pumps using KAN.

The simulations do not account for potential dielectric breakdowns or fluid ionization limits.

How did you ensure the convergence and accuracy of your finite element simulations?

Author Response

Q1: The paper relies entirely on simulations. Where is your contribution?

A: We thank the reviewer for this very relevant comment. We acknowledge that the investigation on influence factors for our photo-induced PLZT-based ion drag pump is based on simulations, but would like to emphasize that this novel photo-induced ion drag pump is firstly proposed in our submitted paper, which enables the fluid pumping via the light. In this way, using the PLZT ceramic patch as the energy source instead of bulky power device can easily realize the miniaturization and integration of ion drag pumps. Furthermore, in order to study the pumping performance, we have constructed an electrohydrodynamics model for the photo-induced ion drag pump as shown in section 2 in the submitted paper. Based on the constructed EHD model, a series of finite element simulations are carried out to investigate influence factors of the pumping performance for the proposed pump, which can be the design guideline for the ion drag pump. The above mentions are our contributions. Finally, we would like to illustrate that some researchers, such as reference 24 and 26, also investigated the EHD mechanism through pure simulations.

Q2: You need provide the real photo of the pump in experiment and experiment setup environment.

A: Thanks for this helpful suggestion. In the submitted paper, we focus on investigations on influence factors of the pumping performance for this proposed ion drag pump via finite element simulations. Following the reviewer's advice, we will carry out a lot of experiments to verify simulation results in our future work.

Q3: You explored several configurations for electrodes. How did you decide the optimal gap and number of electrode pairs? Could there be configurations that you have not yet tested?

A: We thank the reviewer for this very relevant comment. In this paper, we investigated the effect of different electrode structures on pumping performance, which aims to provide a design guideline for the pump instead of seeking an optimal structure. By the control variates method, the effects of different electrode configurations (shown in table 1 in the submitted paper) on pumping performance are systematically studied, such as the gap between positive and negative electrode, gap between electrode pairs, and number of electrode pairs, Besides, the effect of the fluid channel and fluid viscosity are also tested via finite element simulations.

Q4: The manuscript lacks an in-depth investigation into temperature effects and material degradation of PLZT ceramics under UV light.

A: We thank the reviewer for bringing out this important point. As for temperature effects and material degradation of PLZT ceramic, we have detailly studied the effect of temperature caused by the photothermal effect on the output characteristics of PLZT ceramics in our previous research [1]. The results show that, due to temperature effect, a pyroelectric voltage generates between the two electrodes of PLZT ceramic, as shown in equation 1 (where τθ is the time constant of pyroelectric effect). As for the material degradation of PLZT ceramics under UV light, the output performance of photovoltage for PLZT ceramic is stable according to experimental data from studies in recent years. Moreover, degradation and device lifetime issues could be a research point for us to consider in the future.

A: Thanks for this helpful suggestion. As detailed in the first answer, our biggest achievement is the introduction of photo-induced PLZT-based ion drag pumps and the study of their pumping performance. As for the predictive modeling of flexible EHD pumps using Traditional Reqression and Deep Learning Approaches, we think that combining intelligent algorithms and our photo-induced EHD pump is a good advice and research topic, which can be our future work.

Q6: The simulations do not account for potential dielectric breakdowns or fluid ionization limits.

A: We acknowledge that potential dielectric breakdowns or fluid ionization limits are ignored in our simulations. That is because, firstly, our photo-induced PLZT-based ion drag pumps are actually designed using a sandwich-like multilayer structures (i.e., cover layer, channel layer and substrate layer), which are all dielectric materials. The breakdown electric field for our dielectric materials in the paper is about 12 kV/mm, but the maximum working electric field is less than 10 kV/mm. Secondly, the breakdown electric field of dielectric fluid is about 15 kV/mm. Consequently, our photo-induced ion drag pumps can fully avoid the dielectric breakdown.

Q7: How did you ensure the convergence and accuracy of your finite element simulations?

A: In order to ensure the convergence and accuracy of our finite element simulations, several measures are conducted in the submitted paper. Firstly, simulations in our work are all based on the constructed EHD model and boundary conditions set in the submitted paper. Moreover, different meshing schemes are used to simulate the EHD pump and simulation results almost show the same pumping performance, which can further verify the convergence and accuracy of our simulations.

  1. Huang, J.H.; Wang, X.J.; Wang, J. A Mathematical Model for Predicting Photo-Induced Voltage and Photostriction of PLZT with Coupled Multi-Physics Fields and Its Application. Smart Mater. Struct. 2016, 25, 025002, doi:10.1088/0964-1726/25/2/025002.

Round 2

Reviewer 3 Report

Comments and Suggestions for Authors

The paper has been improved